# Management of Donkeys in Assisted Interventions: A Snapshot

**DOI:** 10.3390/ani14050670

**Published:** 2024-02-21

**Authors:** Lucia Sobrero, Emanuela Dalla Costa, Michela Minero

**Affiliations:** Dipartimento di Medicina Veterinaria e Scienze Animali, Università degli Studi di Milano, Via dell’Università 6, 26900 Lodi, Italy; emanuela.dallacosta@unimi.it (E.D.C.); michela.minero@unimi.it (M.M.)

**Keywords:** animal care, donkey welfare, pet therapy, donkey-assisted interventions

## Abstract

**Simple Summary:**

The donkey, along with the dog and the horse, is one of the species most involved in Animal-Assisted Interventions (AAIs). The stoic nature of this species, combined with the working requirements of Animal-Assisted Interventions, mean that the welfare of donkeys involved in such activities needs to be monitored and protected. This article reports information about the management of donkeys housed in six different facilities with varying degrees of experience in Donkey-Assisted Interventions in Northern Italy and emphasizes the preventive value of proper animal management in safeguarding donkey’s welfare.

**Abstract:**

People working in the field of Animal-Assisted Interventions (AAIs) often state that they perceive animal welfare as a matter of paramount importance; nevertheless, most scientific literature focuses on the effectiveness of interventions from the user’s perspective. Before focusing on the animals’ management and welfare during their interactions with users, it is important to ensure animal welfare during their “ordinary lives”. This article reports information and considerations about the management of donkeys involved in AAIs in Northern Italy. Six facilities with several years of experience in Donkey-Assisted Interventions were visited for the purpose of an initial data collection regarding the management of donkeys involved in AAIs. Some knowledge gaps regarding the nutritional needs of the donkey and its preventive medicine have been identified; this study also highlighted a need for efforts to create a more stimulating and enriched environment for animals involved in AAIs. Some possible areas for improvement in the management of donkeys involved in AAIs have been highlighted. Further studies are necessary to provide a more comprehensive picture of the welfare of donkeys involved in this context.

## 1. Introduction

Donkeys are undoubtedly one of the most versatile domestic species, ranging from being a quintessential working animal to a beloved companion. Their success and global spread are deeply connected to their adaptability and capacity to withstand harsh working conditions, even with limited resources [1]. Still today, in various geographical areas around the world, the sustenance of hundreds of millions of people depends significantly on the availability of this animal species, which provides not only work assistance but also food products such as meat and milk [2]. Donkey milk is of interest in high-income countries as well, where it is used not only in the food sector, but also in the medical and cosmetic fields, thanks to its anti-inflammatory and antioxidant properties [3,4].

Recently, in many contexts, donkeys have acquired new significance, moving from the role of livestock or a working animal to that of a relational subject, thus also becoming appreciated for their unique relationship with humans. Together with the dog and the horse, it is one of the most involved species in Animal-Assisted Interventions (AAIs) [5].

In Italy, the National Guidelines for AAIs define three different types of interventions, which vary in terms of objectives, number of users, and types of professional figures involved. Animal-Assisted Activities (AAAs) represent the simplest type, with a playful–recreational group meaning. Animal-Assisted Educations (AAEs) are interventions with educational value, either individual or group-based. Lastly, Animal-Assisted Therapies (AATs) are highly structured individual interventions with specific therapeutic objectives [6].

Donkey-Assisted Interventions (DAIs) represent one of the possible variations of what is known as Social Farming or, in a broader sense, Greencare [7]. Their common goal is to enhance the quality of life for individuals, whether with or without vulnerabilities, through contact and care of donkeys, within a typical rural setting [8,9]. Activities typically serve recreational purposes, falling within the category of AAAs, due to the lower costs and the involvement of a limited number of professional figures [10]. Many authors report that, to date, the donkey is involved in activities that only partially overlap with those carried out by horses. Referential and care-related activities are proposed for both species; however, in the case of horses, riding work often plays a central role, given its motor, balance, and proprioceptive benefits. Conversely, with donkeys, ground-based relational work is typically preferred [7,11].

Despite numerous past and present stories of mistreatment, this species seems to be finally valued in the context of Assisted Interventions [1,12]. Nevertheless, the Italian National Guidelines emphasize the “work-oriented” nature of AAIs [6]. Other sources highlight the need to define at least the voluntary or professional role of animals involved in such a context [13]. Since AAIs involve not only physical, but also mental and emotional engagement, activities with the users of the animal have to be considered work.

Thus, before focusing on animal management during interactions with patients, it is important to ensure animal welfare during their “ordinary lives” as well, both in periods of rest and of work outside activity hours. This need is justifiable from both a practical and an ethical perspective. In the field of animal husbandry, it is widely accepted that optimizing management means preventing a range of health and behavioral problems that can impact not only animal welfare, but also the economic well-being of farmers. This issue is extensively described regarding various livestock species [14,15,16,17], including, more recently, donkeys [18]. This should also apply to AAIs: if an animal represents a means to improve users’ quality of life, it should enjoy ideal conditions to lead a good existence in relation to the fulfillment of its needs. As suggested by Fine and Griffin, animal welfare should be understood not only as a means of protecting the animal, but also as crucial to the successful delivery of patient care services [19]. Moreover, one of the most ambitious theoretical goals of Assisted Interventions is to embody the concept of One Welfare, the idea of a deep interconnection between human well-being, animal welfare, and the environment [20,21]. In the context of AAIs, the One Welfare perspective would imply that the user’s welfare cannot be considered more important than the animal’s. This need can easily become a critical point, especially for those species that, due to their evolutionary history, manifest conditions of discomfort subtly. Among domestic animals, the donkey presents a challenge in that it often expresses fear, pain and illness with mild signs that correspond to subtle behavioral changes, hardly perceptible to an untrained eye [22,23]. This aspect, at least in theory, would make it necessary to pay special attention to prevention in management, starting from basic aspects such as where the animals live, to how they are fed, and how their healthcare is managed, as several authors report [24]. Regarding milk donkeys’ management, Dai and colleagues observed a significant heterogeneity in how they are fed, housed, and managed from a health perspective in Europe. In response to this, they formulated guidelines, defining best practices based on the existing literature [18]. There are currently no specific indications regarding donkeys involved in educational farms or in Animal-Assisted Interventions.

Given this premise, the aim of this study was to conduct an exploratory data collection concerning the management of donkeys involved in DAIs facilities located in Northern Italy. This approach aims to be an illustrative step, preliminary to the collection of data on a much larger scale, leading to the subsequent definition of best practices to enhance the quality of life for animals involved in this field.

## 2. Materials and Methods

### 2.1. Donkey Facilities

Six facilities that regularly carry out Donkey-Assisted Interventions were recruited on a voluntary basis; the sample represented the result of an ad hoc sampling, suitable for exploratory projects. Data were collected between July 2022 and April 2023 through on-site visits and interviews with the facility managers. All the recruited facilities were located in Northern Italy and had several years of experience in the field of DAIs.

### 2.2. Survey Method

Given the heterogeneity of the contexts and the limited literature on the subject, information was collected in a semi-structured manner, using a closed-ended questionnaire (yes/no answers) and through open conversation with the facility manager focused on the day-to-day management of the donkeys involved in DAIs. The questionnaire consisted of four sections, the first encompassing general information regarding facilities, personnel, and donkeys; the other three sections were focused on housing, nutritional management, and preventive healthcare procedures, as presented in Appendix A. The questionnaires were filled out on-site during the interview with the facility manager. In only one instance, the interview was conducted remotely, and therefore, it was not possible to physically visit the facility. After completing the questionnaire, data on donkeys’ identifications (age, sex, and final destination of the animal, in the sense of the possibility or impossibility of being used for food production) were collected.

### 2.3. Data Analysis

Data obtained from the questionnaire were reported in an Excel file and then analyzed with descriptive statistics (percentage of answers for each considered section). For the sections entitled Housing, Feeding, and Health, graphs were prepared using Excel to facilitate an overview of the results.

## 3. Results

### 3.1. Facilities, Personnel, and Donkeys Involved in AAIs

All the recruited facilities were in Northern Italy and managed by non-profit organizations. A minority of them (17%) was recognized as a Specialized Center in Animal-Assisted Education and Therapy, having undergone an inspection by the local health authority to verify the requirements specified in the national guidelines. In terms of activities, 67% of the facilities organized exclusively recreational interventions (AAAs), 33% conducted all types recognized by National Guidelines (AAAs, AAEs, and AATs). Regarding the staff, in 67% of the facilities, personnel were trained for AAIs, while 33% held either a degree in Animal Husbandry and Animal Welfare or a master’s degree in Veterinary Medicine. Half of the facilities did not collaborate with any veterinarian trained for AAIs and consulted a clinical veterinarian only when necessary; in the other facilities, a veterinarian trained in AAIs was present on-site more (33%) or less (17%) frequently then once a month. In most of the facilities (83%), the feeding and cleaning operations were carried out by the same individuals who organized the AAIs. A groom responsible exclusively for these operations was present in 17% of the facilities. The considered facilities hosted from 3 to 11 donkeys, mostly non-purebred (83%), aged between 1 and 26 years old (mean: 13.3 years; median: 14 years; SD: 5.9). Out of a total of 34 subjects, 44% were males and 56% were females, as presented in Table 1. Most of the males were castrated (94%), except for two due to their young age. Most of the donkeys were registered in the National Database as Not Destined for Food Production Animals or NDPAs (91%); a small number of them were registered as Destined for Food Production Animals or DPAs (9%).

### 3.2. Housing

In the considered facilities, the donkeys lived in groups ranging from three to eight animals, in stables that could be categorized into two main types: paddocks with a small shelter (50%) or stables with regulated access to a paddock (50%). The shelter and the stables exhibited significant structural differences, ranging from structures built by specialized companies (50%) to home-made solutions (50%); in all cases, the resting areas had a concrete floor covered with bedding. Regarding the features considered for the paddocks and their relative percentage are reported in Figure 1. Sixty-seven percent of the facilities had grass-covered paddocks (as opposed to 33% without grass); In 33% of the cases, the paddock was accessible to the animals throughout the year; in the remaining 33%, access was regulated based on weather conditions or season. In 33% of cases, spontaneous shrubs were also present in the paddock. Half of the facilities reported that the paddock had a stable ground, without a tendency to become waterlogged (as opposed to 50% with unstable ground). Shaded areas different from the shelter were present in 50% of the facilities (as opposed to 50% without shade). In most of the contexts observed (83%), environmental enrichments, in the sense of something intentionally added to the environment to make it more stimulating for the animals, were absent at the time of the visit. Only one facility claimed to pay particular attention to this point, organizing weekly enrichment plans and rotating them over time. However, this information was obtained through a remote interview, and therefore, in this specific case, a contextual evaluation was not possible.

### 3.3. Feeding

In most facilities, donkeys were primarily fed with hay (83%); the remaining facilities used straw as the primary forage, with small addition of hay proportioned to the weight of the donkeys (17%). In addition to forage, all facilities included one or more supplements, as shown in Figure 2. Permanent pasture was available in 33% of the facilities. As occasional supplements, once a week, edible plant shrubs (17%), fruits and vegetables servings (50%), and cereal-based feeds (33%) were included in the diet. Most of the facilities (83%) used fruits and vegetables as rewards during activities involving users. Regarding the forage administration, in 83% of the facilities the forage was rationed and distributed in two meals. In one facility (17%), it was provided ad libitum, always made available to the animals. The forage was distributed to the animals in a feeding trough (67%), inside suspended nets (33%), or scattered on the ground in the paddock (50%), as illustrated in Figure 3.

### 3.4. Preventive Healthcare Procedures

The healthcare procedures in the considered sample are summarized in Table 2. These include both routine operations carried out by non-veterinary trained personnel, such as weight monitoring and hoof care, as well as veterinary procedures relevant for prevention purposes, in agreement with a renowned text on donkey clinical practice [25]. In Figure 4, the percentages of such procedures adopted by the facilities are reported. The monitoring of weight and annual dental examinations were performed in 17% and 33% of the facilities, respectively. All the recruited facilities vaccinated the donkeys annually against influenza and tetanus. Only 33% of them, in addition to the basic vaccination, also vaccinated for West Nile disease. Half of the facilities controlled endoparasites by conducting fecal examinations and selectively treating the animals once a year; the other half blindly administered treatments once or twice a year.

## 4. Discussion

The aim of this study was to collect exploratory data concerning the management of donkeys involved in DAIs in Northern Italy. The gathered information has served as a starting point for the discussion and for considerations, based on the existing literature about the optimal management of donkeys in the context of Animal Welfare Science. Only a minority of the facilities considered was a Specialized Center in AAIs. In Italy, all the AAI facilities are authorized by the Local Health Authority responsible for the area. Despite the National Guidelines for AAI making a distinction between Specialized Centers and unrecognized facilities, both types of structures can provide all types of interventions, including activities with educational (AAE) and therapeutic (AAT) purposes. Overall, 67% of the facilities recruited organized only recreational activities (AAAs), while 33% of them conducted all types of interventions. These results fit well with the fact that all the structures considered were managed by non-profit organizations. In fact, it is well known that organizing AAAs is often a way to contain costs, as they typically involve a limited number of professional figures. Our findings align with the general trend reported by other authors [5,26].

Regarding the personnel, a heterogeneous picture has emerged, ranging from the exclusive presence of staff trained in AAIs but lacking other qualifications relevant for the welfare of the donkey to the presence of personnel either with a master’s degree in Veterinary Medicine or a degree in Animal Husbandry and Animal Welfare. This is the first study that investigated the personnel’s education related to animal welfare, an important issue because of the well-known link between the quality of the human–animal relationship and animal welfare [27]. However, countless factors, both related to the animal and to the handlers, can influence the quality of this relationship and, consequently, the animal’s welfare state [28,29]. Only a minority of the facilities had frequent collaboration with a veterinarian trained in AAIs, with an on-site presence of more than once a month; in the remaining facilities, such collaboration was either absent or sporadic, occurring less than once a month. Both guidelines and the scientific literature identify interdisciplinarity as the key to the success of AAIs as well as for the protection of animal welfare. The concept of One Welfare itself encompasses the idea of interdisciplinarity [12,30]. However, both our findings and the existing literature seem to reflect a certain challenge in realizing this principle. Galardi and colleagues (2022) attributed the limitations faced by AAI providers to the lack of funding and to the absence of a network within the national health system [26]. As, in most cases, establishing a truly multidisciplinary team is not feasible due to cost constraints, it would be beneficial to explore alternative approaches that allow for some level of integration of expertise without excessively burdening costs.

Regarding the donkeys, the facilities recruited hosted from 3 to 11 subjects, with some gender uniformity but a highly variable age range. When the Standard Deviation is high, it is typically due to a large variation in values within the sample. Indeed, among the donkeys, there were very young subjects (1–2 years old) and elderly subjects. In this regard, it is important to note that the National Guidelines discourage the use of animals that are too young or too old in AAI programs [6].

### 4.1. Housing: From Mere Containment to Dynamic Context

In the recruited facilities, donkeys were housed in paddocks with a small shelter or in a stable with regulated access to a paddock, that were represented by both home-made solutions and supplies provided by specialized companies in animal supplies. It is well known that housing systems represent an important, although not exhaustive, aspect of animal management, as they can significantly impact their welfare [31]. Since national guidelines do not provide accurate indications regarding equine housing systems, some Italian regions have published more detailed documents, encompassing both structural and management aspects [32]. The above reported documents and guidelines highlight that facilities must meet not only physiological, but also ethological needs; however, information related to the implementation of the latter is still lacking. In our study, some factors that could affect both physical health and behavior have been investigated. First, the presence of pasture can be relevant from both a nutritional and behavioral standpoint, as it ensures the intake of macro- and micronutrients and increases movement during feeding [33]. More than half of the facilities presented grass-covered paddocks, but only in a minority of cases were they accessible to donkeys permanently throughout the year, while in the others, access was regulated on a seasonal basis. In this regard, it is important to remember that providing small portions of grass progressively can prevent a sudden intake of highly fermentable substances, which may lead to health issues [34]. Moreover, careful pasture management to limit excessive trampling is essential to avoiding the progressive depletion of the pasture [35]. Given that the donkey, unlike the horse, is both a grazing and browsing herbivore, the presence of spontaneous shrubs in the living environment was investigated. Our investigation revealed that only a minority of the facilities had such shrubs. Half of the facilities highlighted soil type as a critical issue due to its tendency to become waterlogged, muddy, and impractical for the animals. From a health perspective, this aspect is particularly relevant in relation to the characteristics of the donkey’s hoof, which, compared to that of a horse, has a greater tendency to absorb and retain water; in conditions of excessive moisture, hoof pathology is more likely to occur [36]. Half of the facilities did not have shaded areas other than those provided by the shelter or by the stable, which could be a critical aspect, especially during summer when animals tend to seek shade as temperatures rise [37]. One of the most surprising findings of the study is that nearly all the facilities visited did not present environmental enrichments within the living environment of the animals at the time of the visit. Only 17% of them claimed to pay particular attention to this aspect, organizing weekly enrichment plans and rotating them over time. However, this information was obtained through a remote interview, and therefore, in this specific case, a contextual evaluation was not possible. It is well known that an animal living in a poorly stimulating environment will exhibit less interest in the environment itself, which may translate into reduced exploratory behavior, apathy, or a diminished responsiveness to surrounding stimuli [38]. In the case of horses, states of anhedonia with significant cognitive and affective impairments have been described as an extreme consequence of confinement in deprived environments [39,40]. Such a scenario not only represents an animal welfare issue, but also clearly contradicts the purposes of AAIs and the related One Welfare principle. Although there are currently no available studies on donkeys, it is reasonable to assume that a poorly stimulating environment might affect the animal’s interest in proposed activities or its motivation for interaction with handlers and users. 

This consideration should prompt those who hold animals for AAI purposes to focus on prevention, actively engaging in transforming their living environment from a mere container to a dynamic context that stimulates the animal’s physical, mental, and emotional activity. Knowing the ethogram of a species allows for planning an environment sufficiently stimulating for the animals, thus predisposing them to engage in species-specific behaviors relevant to their physical, mental, and emotional well-being [41,42].

### 4.2. Feeding: Managing Nutrition from Both a Nutritional and Behavioral Perspective

As the donkey is a strict herbivore, the primary forage was investigated and it was found that most facilities fed the animals only hay, while the remaining facilities used straw as dietary basis with small supplements of hay. This result is quite surprising and deserves further exploration; indeed, donkeys, having evolved in semi-arid environments, exhibit specific physical and metabolic traits that make them excellent utilizers of highly fibrous and energetically poor foods. For these reasons, barley or wheat straw, besides offering a clear economic advantage, is more suitable for the metabolic needs of donkeys, and when provided freely to the animals, it allows for an extended feeding time without predisposing them to weight gain [43,44]. In addition to the forage, all the facilities provided animals with one or more supplements, such as permanent access to grass-covered pasture, edible plant shrubs, fruit and vegetable servings, and cereals. In horses, it has been shown that varying the type of fiber has positive effects on foraging behavior, which encompasses all aspects of feeding behavior before ingestion, such as olfactory exploration, grasping and manipulation with the lips, and chewing [45]. Although there are currently no similar studies on donkeys, it is reasonable to believe that they could also benefit behaviorally from a variation in fiber type, without significantly impacting the diet’s energy intake. In support of this assumption, it is important to note the complexity and variability of foraging behavior among donkeys kept in the wild in different geographical areas [46,47]. Moreover, some evidence suggests that the combination of grazing and foraging on shrubs contributes to containing parasitic burdens [33]. The administration of grains or cereal-based feeds has been investigated for potential negative health consequences. It is well recognized that donkeys should not consume feed excessively rich in starch and simple sugars, as an excess of energy in this species easily predisposes them to obesity and related metabolic alterations [48]. For this reason, fruit too should not be included in the diet in large quantities. In most facilities, fruits and vegetables were used as rewards for the animals during activities with users; however, half of them also integrated them into the diet of the donkeys on a weekly basis. In the context of Animal-Assisted Interventions, aiming to maximize animal welfare, supplements could be used strategically: they can be used, for example, to introduce novelties into the living environment of animals. Alternatively, considering the unique abilities of the donkey species, they can be utilized to set up games with a cognitive component or employed as rewards to teach beneficial behaviors with a Positive Welfare approach [49,50]. As nutrition is not just about what, but also about how food is consumed, information regarding the method of administering feed has been investigated. In most of the facilities, the forage was rationed and distributed in two meals; only in a few cases was it provided ad libitum, or in other words, always made available to the animals. When rationed, the forage was distributed to the animals in feeders, inside suspended nets or scattered on the ground in the paddock. In its natural conditions, the donkey spends 14–16 h feeding, moving over long distances; however, in most contexts, as observed in the present study, animals remain stationary at the feeder for the duration of the meal, which often occurs rapidly during food distribution, causing potential fluctuations in gastric pH and blood insulin levels. For these reasons, the practice of feeding these animals in a meal-like manner should be discouraged [34,51]. 

Regarding nutrition, this study highlights some gaps in knowledge concerning the nutritional needs of the donkey species. Despite the gathered information seems to confirm the general tendency to overfeed donkeys, drawing conclusions without an assessment of the nutritional status of the animals is imprudent [43,52]. Therefore, further studies conducted across a greater number of DAI facilities and incorporating animal-based indicators are necessary to confirm our assumptions.

### 4.3. Preventive Healthcare Procedures: Let Us Make Prevention a Keyword

The last section of the questionnaire aimed to evaluate the implementation of donkeys’ healthcare operations in the recruited facilities. The health of an individual is an integral, albeit not exhaustive, part of animal welfare. Broom in 2006 defined it as an individual’s attempt to cope with pathology [53]. Among domestic species, the donkey is known to exhibit pain, stress, and illness with mild signs, often not visible to untrained eyes; therefore, health prevention deserves special attention, as many authors have highlighted [24,54]. Weight loss, in some cases, can be the only sign of a health problem [48]. Thus, monitoring the weight of the animals is a useful practice to identify early weight loss and, concurrently, to prevent the negative consequences of overfeeding. In the recruited facilities, this practice was routinely applied in only a minority of cases. Donkeys, like horses, have continuously growing teeth, making it good practice to subject animals to an annual dental examination by a veterinarian, preferably a qualified equine dentist. Furthermore, while in horses a dental issue typically manifests with an immediate interruption of food intake along with other symptoms, donkeys tend to lose their appetite only in advanced stages of the pathology; therefore, prevention is important also in this case [24]. In our study, less than half of the facilities subjected donkeys to annual dental check-ups.

Another relevant aspect in terms of prevention in equines is hoof management. Given that a donkey’s hoof should be trimmed approximately once every 6 to 10 weeks [25], an inquiry into the percentage of facilities implementing this practice once every three months or more often showed that more than half followed this frequency. However, it is important to note that trimming frequency should be correlated with various factors, including aspects related to the animal (such as foot conformation and the degree of hoof wear) and the environment (for example, the type of terrain). Good hoof management, carried out by an experienced trimmer, is therefore important in preventing many pathologies that cause lameness [55]. Regarding the prevention of infectious diseases, all the facilities vaccinated for Equine Influenza and Tetanus, with only a minority of them vaccinating for West Nile Disease also. Concerning basic vaccination, currently in Italy there are no commercially available monovalent vaccines for individual pathologies [56]. One possible consequence is that donkeys may be excessively vaccinated for tetanus; further studies would be necessary to investigate the potential health implications of this fact. West Nile Disease is a zoonosis transmitted by mosquitoes. Equids, like humans, serve as accidental hosts, playing a marginal role in viral transmission. Nevertheless, in horses, as well as in donkeys, sporadic cases of neurological forms, some of which can be fatal, have been reported in Europe [57]. Finally, regarding the control of endoparasites, half of the facilities claim that they annually conduct fecal exams and treat animals only when necessary; in the remaining contexts, treatments were blindly administered with a frequency of once or twice a year. However, the increasing phenomenon of anthelmintic resistance (AHR) in equines’ parasites suggests the need for a different approach. From this perspective, strategic deworming based on fecal worm egg count (FWECs), along with proper pasture management involving feces removal, represent fundamental elements for prevention [24,58]. 

The information collected for this study, although related to a limited number of contexts, suggests a certain heterogeneity in practices related to donkey preventive medicine. Further studies conducted on a larger number of facilities would be necessary to assess any correlation between a greater presence of veterinarians and the quality of healthcare provided to the animals.

## 5. Conclusions

Starting from the visit of six Donkey-Assisted Intervention facilities in Northern Italy and the related routine management of the donkeys, this study aims to represent a first step towards greater considerations into the preventive value of optimal animal management in the context of AAIs. Some possible areas for improvement in the management of donkeys involved in AAIs have been highlighted. Some deficiencies of knowledge regarding the nutritional needs of the donkey and its preventive medicine have been identified. The study also highlights a need for efforts to create a more stimulating and enriched environment for the animals. The authors believe that the case of the donkey, a species typically mistreated regardless of its use, can be well-suited to stimulate a heightened awareness of what the One Welfare approach would demand to fulfill. Further studies are necessary to provide a more comprehensive picture of the welfare of donkeys involved in this context.

## Figures and Tables

**Figure 1 animals-14-00670-f001:**
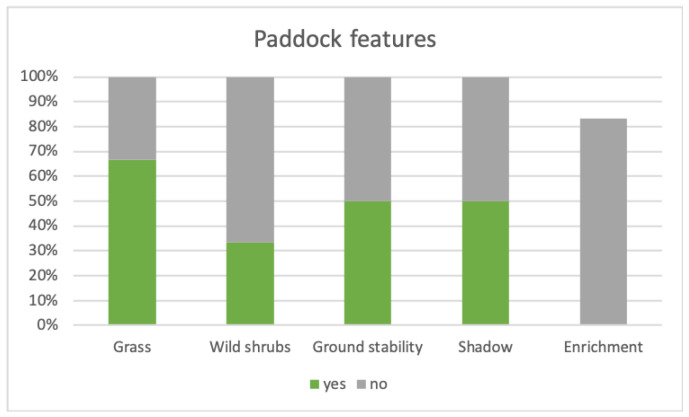
The graph reports the percentages of DAI facilities that implemented some paddock features reported to be relevant from both a physical and a behavioral perspective.

**Figure 2 animals-14-00670-f002:**
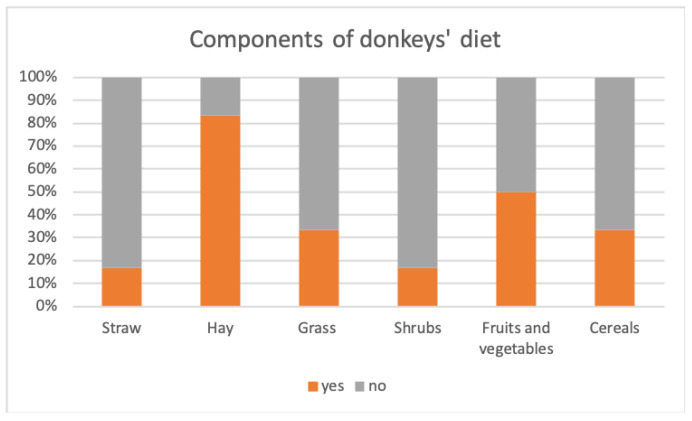
The graph reports the percentages of DAI facilities that fed donkeys with the considered components of the diet.

**Figure 3 animals-14-00670-f003:**
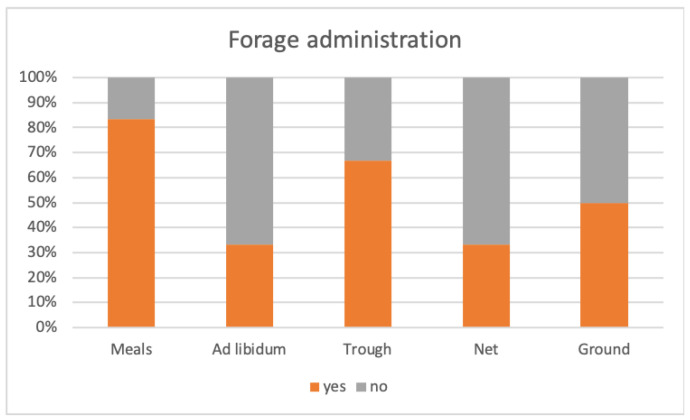
The graph reports the percentages related to different methods of administering forage in six DAI facilities in Northern Italy.

**Figure 4 animals-14-00670-f004:**
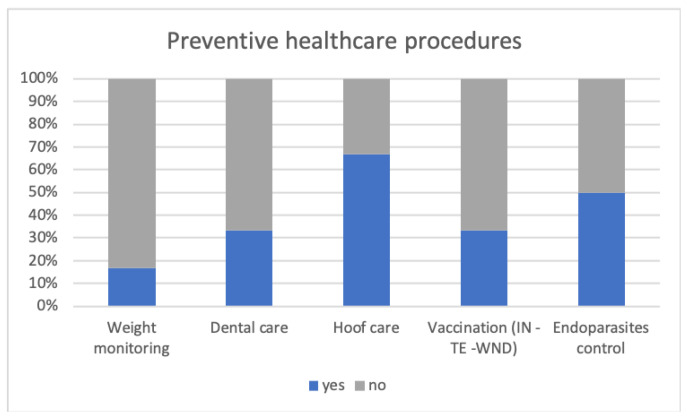
The graph reports the percentages of the preventive healthcare procedures adopted by six DA facilities in Northern Italy.

**Table 1 animals-14-00670-t001:** Information about the DAI facilities and the donkeys involved in Assisted Interventions.

ID	Location of the Facility	Number of Donkeys per Facility	Number of Males	Number of Females	Average Age per Facility	Final Destination ^1^
1	Piemonte	8	7	1	15.4	NDPA
2	Veneto	4	0	4	18.5	NDPA
3	Trentino	11	3	8	15.5	NDPA
4	Lombardia	4	1	3	6.3	NDPA
5	Lombardia	3	1	2	6.3	DPA
6	Lombardia	4	3	1	10.3	NDPA

^1^ DPA and NDPA: Italian acronyms for animals destined or not destined for food production, respectively.

**Table 2 animals-14-00670-t002:** Donkeys’ healthcare procedures investigated in the six DAI facilities considered.

Procedure	Description
Weight monitoring	Weight measurement or estimation through chest circumference and height at the withers
Dental care	Dental check-up and corrective interventions
Hoof care	Hoof check and potential trimming
Vaccinations	Vaccinations for Equine Influenza, Tetanus, and West Nile Disease
Fecal exam	Fecal exam and selective treatment (vs. blind administration of anthelmintic drug)

## Data Availability

Data are contained within the article and Appendix A.

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
