# Peer review of "Management of Donkeys in Assisted Interventions: A Snapshot"

_animals, 2024, doi:10.3390/ani14050670_

Round 1

Reviewer 1 Report

Comments and Suggestions for Authors

The paper described the management of donkeys in assisted interventions scene in Northern Italy. However, this would better give some much needed context for your research.

General Questions

1. When you refer to donkey facilities, are these sanctioned, if so, with whom?

2. The methods section does not provide enough detail and clarity on your enrolment process of study. Whether the six donkey facilities could represent or reflect the overall donkey facilities population in Italy?? I have some concerns about the internal validity of your survey, meaning, how did the managers get information on all the donkeys in their facilities? Who owns and makes decisions about the facilities donkeys, the managers, or the government? This is important information to guide donkey feeding and health care. 

3. This would better give some much needed context for your research. For example, how are donkeys exercised, and used in these facilities? These may be associated with donkey feeding and health care.

Specifics

1. Line 28: Keywords: delete “animal welfare” and “animal-assisted interventions”, replaced by another two words not shown in the title.

2. Line 44: What’s the meaning of abbreviation “AAI”?

3. Introduction section: The first paragraph is too long. Please divide it into several shorter paragraphs for clarity.

4. Line 99: “Donkey-Assisted Interventions (DAI)” should be replaced by “DAI”.

5. Line 112: What’s the meaning of “not productive fate”? Please rephrase.

6. Line 115-118: The data analysis is too simple, please describe standard deviation (SD).

7. Line 126-127: abbreviation of “AAA, AAE, AAT” should be given the clear definition throughout the paper.

8. Line 121-141: the basic natural information of donkey facilities (geographical location, size, number of workers, etc.) and donkeys (numbers, gender, age, breed, etc.) should be displayed in a table.

9. Line 165: What’s the meaning of “enrichment”?

10. Line 189: “3.5” should be replaced by “3.4”.

11. Please give more results of preventive healthcare procedures, such as frequency of dental care, hoof care and fecal exam per year.

I do not consider the paper ready for publication until these issues are addressed. Having said that I look forward to reading future manuscripts.

Author Response

Generals

1.      When you refer to donkey facilities, are these sanctioned, if so, with whom?

In Italy donkey facilities are authorized by the Local Health Authority responsible for the area. The AAI National Guidelines make a distinction between Specialized Centers and unrecognized facilities, but both types of structures can provide all three types of interventions (AAA, AAE and AAT). In the case of facilities not recognized for AAI, they undergo inspections like any other facilities where donkeys are housed. In the case of Specialized Centers higher requirements should be verified, as detailed in documents such as the one mentioned in the article at line 272. Nevertheless, these documents vary by region and typically do not provide practical indications regarding the nutrition and the health management of the animals, nor on how to meet their ethological needs.

2.      The methods section does not provide enough detail and clarity on your enrolment process of study. Whether the six donkey facilities could represent or reflect the overall donkey facilities population in Italy?? I have some concerns about the internal validity of your survey, meaning, how did the managers get information on all the donkeys in their facilities? Who owns and makes decisions about the facilities donkeys, the managers, or the government? This is important information to guide donkey feeding and health care.

Comments regarding the representativeness have been intentionally omitted. In Italy there is a database where all AAI facilities (both Specialized Centers and unrecognized structures) should be registered. Nevertheless, a significant number of facilities, despite engaging in AAI, are absent from the database. This is probably due to the absence of a binding regulation on the matter. For this reason, it is rather challenging to establish with certainty the representativeness of the results. However, among the facilities recruited, there was a well-known Specialized Center for AAI and an internationally renowned no-profit organization dedicated to the protections of donkeys and to the education on their welfare. Could this be a significant aspect to mention?

In most of the facilities recruited, the manager was the same person responsible for managing the donkeys and organizing assisted activities with users. In other cases, the person in charge of animal activities was directly involved in the questionnaire completion process to ensure the completeness and accuracy of the collected information.

Do you think it could be helpful to include this information in the text in the ‘Survey Method’ section?

3.      This would better give some much needed context for your research. For example, how are donkeys exercised, and used in these facilities? These may be associated with donkey feeding and health care.

Donkey preparation was intentionally excluded from the study as it is, in our opinion, a broad topic deserving of specific research. Nevertheless, facility managers were informally questioned about the training and the workload of the donkeys (not through the questionnaire). The variability in practices was extensive, ranging from animals never trained to some that have received training from individuals with varying levels of expertise; and ranging from animals completely at rest in winter and engaged in daily work in spring and summer to animals working more or less continuously throughout the year. This has strengthened our belief that the training is a topic deserving a separate in-depth exploration.

Specifics

1.      I have replaced “animal welfare” with “animal care” and “animal-assisted interventions” with “pet therapy”.

2.      Line 45: I have replaced the abbreviation “AAI” with “Animal-Assisted Interventions”.

3.      As suggested, I have divided the first paragraph of the introduction into shorter paragraphs for clarity.

4.      Line 99: as you suggested, I have replaced “Donkey-Assisted Interventions” with the abbreviation “DAI”. Please see Line 111.

5.      I have replaced “productive/non-productive fate” with “the final destination of the animal”, indicating its meaning in accordance with Italian law. Please see Line 125.

6.      I have included a comment regarding the standard deviation (SD) in the first paragraph of the discussion (line 260). Additionally, I have included the average age average per farm in the table containing information about facilities and donkeys (Table 1, line 155). 

7.      As requested, I have included the definitions of AAA, AAE and AAT in the introduction. Please see line 46-51.

8.      As suggested, I have inserted a table containing the general information about facilities and donkeys (Table 1, Line 155).

9.      I have clarified the meaning of enrichment in the text. Please see line 174.

10.  I have replaced “3.5” with “3.4”.

11.  I have specified more, including in the results (line 210, 214), as well as in the discussion (Line 382, 398), the relative frequencies of the healthcare procedures.

Reviewer 2 Report

Comments and Suggestions for Authors

Please state in full what the acronym AAA, AAE and AAT are the first time they are used.

This is a very interesting paper, providing scope for further work

There are a few minor areas in which syntax could be improved and a native English speaker could improve. These do not detract from the comprehension or merit of the paper.

Comments on the Quality of English Language

There are a few minor areas in which syntax could be improved and a native English speaker could improve. These do not detract from the comprehension or merit of the paper.

Reviewer 3 Report

Comments and Suggestions for Authors

I must congratulate the authors because the manuscript seems necessary for the fate of such a beloved animal as the donkey. So, thank you for showing us that possibility and doing it with originality and know-how. For example, the allusion to how to reward with food during activities is quite interesting and convenient.

I just wanted to suggest tiny changes, or at least, some response to a pair of doubts.

In Methodology, you affirm that six interviews were done, in most cases in person, except on one occasion. In that case, I feel that some information is missing from the proper interviews. I am not sure when reading, whether the interview is a yes-no survey plus open-ended questions; or just a limited survey. The former option implies that during the Results, it could be richer to add certain paragraphs extracted from the interviews, just to clarify or support your points of view in the Discussion.

Another aspect that I feel is missing is the link between some qualities such as infrastructures and buildings to possible and pertinent activities. That way, you could propose some types of activities as long as certain conditions are met. You have done it successfully concerning Feeding, but I kind of miss it in Housing and Health.

Anyway, congratulations for your work, which I see as a good guide to start working with donkeys fairly and carefully.

Author Response

Point-by-point response to Comments and Suggestions for Authors

1. As I stated in the Materials and Methods (section Survey Method), in all cases, including the remote one, the interviews were conducted in a combined manner—partly through the use of a questionnaire and partly through free conversation with the facility manager. For convenience and due to the breadth of topics addressed, the article specifically reports and discusses the questionnaire results. The wealth of additional information collected has primarily served to build a background that can guide the objectives of future research.

2. I'm not entirely sure I understood the second issue raised. In our view, the link between housing and employment in assisted activities lies in this: an environment that is as stimulating as possible (both physically and mentally) and preventive healthcare should be a prerequisite to have animals that are cognitively and physically fit for the job with patients.

Do you think this aspect should be clarified or explicitly elaborated further in “Housing” and “Health” sections?

Reviewer 4 Report

Comments and Suggestions for Authors

Manuscript Review animals-2876923

Overall comment:

This is an interesting article highlighting the importance of general management for donkeys involved in AAI. As this is an exploratory study and no animal-based indicators were assessed I think some more caution needs to be taken in the wording in places to avoid overclaiming. The graphs presented could also do with some amendments.  However, I enjoyed reading the manuscript and I believe that it is relevant to the future of the sector, I have made a few suggestions below:

Summary/Abstract

L8 re-order to ‘species most involved in’

L9-10 This sentence doesn’t currently work – perhaps try something like: ‘The stoic nature of this species, combined with the working requirements of AAI mean that the welfare of donkeys involved in AAI needs to be monitored and protected.’

L17 ‘it would be’ could be changed to ‘it is’

Introduction

L36 ‘certain populations’ this is actually a high number of people, estimated about 600million globally and donkeys can be people’s sole source of income. You could emphasise their importance as working animals a little more here.

L38 ‘industrialised countries’ doesn’t sound great, maybe try high-income countries instead

L75 Remove ‘one’

L80 Remove ‘a’

L83 This could do with a little more context for the reader – heterogeneity in what practices – housing, milking, feeding? Across the industry? In Italy only?

L85 ‘specific indications’ do you mean comparable guidelines like you were talking about for milk donkeys?

L87 amend to ‘the aim of this study was to conduct exploratory data collection’

Materials and Methods

L95 Maybe add a little more context - six facilities that regularly carry out AAI using donkeys?

L115 Maybe recorded rather than reported

Results

L127 Could you explain what AAA, AAE and AAT represent for readers

L137 I’m not sure if the journal require decimal points or commas for the statistics?

L153/153 You say 33% and then remaining 33% - is there one missing or do you mean remaining 66%

L180 Maybe illustrated rather than resumed

Graphs – The graphs could do with a bit of work. They would look better without the lines behind the bars and the scale looks crowded, perhaps it would be possible to put only 20%, 40% etc. Rather than every 10%. You could also just represent the yes element of the answer rather than include the no which completes the bar as it would be easier for the reader to see at a glance the most utilised features.   

Discussion

L207 Re-phrase: was to ‘collect exploratory data concerning’

L210 Replace proper with something like optimum

L211-216 This is repetitive of content already stated in your results – can you summarise it or discuss without directly repeating the results again?

L218 – could you be more explicit here – how does AAA help contain costs?  Is it because AAA is cheap to run but you can make high profit from it?

L220-225 I would be a bit cautious in your wording here, obviously having people with theoretical knowledge on animal health from qualifications is valuable but I would hesitate to draw a direct link between an individual having a qualification and being the best in practical terms with the animals – as you mention there is also a certain amount of experience needed around the animals and the personal qualities someone possesses will influence an individual’s interaction quality.

L233 can you qualify this – are you saying the lack of an AAI trained vet constitutes a lack of interdisciplinarity? It seems a very generalised comment.

L263  ‘Resulting in’ doesn’t make sense, try ‘these were present in’

L270 Could you maybe add what are the typical summer temperatures for the region the facilities were in?

L272 Do you think that it could be a bad thing if the same enrichment was left in the environment all the time as it may lose its novelty and hence interest from the animals?

L279 Amend: ‘In the case of horses, states’

L293 Try amending ‘the primary forage was investigated and it was found that most of the facilities fed the animals only hay’

L332 This could do with a reference

L333-339 This needs revising as currently it comes across as very critical but as the nutritional status of the animals was not assessed you have no evidence to back up your claims. Don’t work on assumptions and try to be more cautious in your wording.

L349 Remove ‘well’

L359 Change hooves management to hoof management

L369 Either here or in your results I think you could explain the significance of vaccination for West Nile virus – is it particularly prevalent in Northern Italy or rare but serious etc.

L383 amend ‘greater presence of veterinarians’ do you mean more frequent routine veterinary assessment? Or the presence of an on-site veterinarian?

Conclusion

L389 remove proper, try optimal instead

L391 – ‘a general lack of knowledge regarding their nutritional needs of the donkey’ I think, given the data, this is overclaiming. You didn’t actually try to assess people’s knowledge, just what they are currently implementing which could be affected by a host of other factors (feasibility in terms of costs, time and practicality). You could say that feeding practices have been identified that could be optimised when taking into account the environment and activity of the donkeys involved in your study. This will need amending in your abstract as well.    

Comments on the Quality of English Language

.

Round 2

Reviewer 1 Report

Comments and Suggestions for Authors

Accept